# Real higher-order Weyl photonic crystal

Yuang Pan[1,2,3,4,7], Chaoxi Cui[5,6,7], Qiaolu Chen[1,2,3,4,7], Fujia Chen[1,2,3,4], Li Zhang[1,2,3,4], Yudong Ren[1,2,3,4], Ning Han[1,2,3,4], Wenhao Li[1,2,3,4], Xinrui Li[1,2,3,4], Zhi-Ming Yu [5,6] ✉, Hongsheng Chen [1,2,3,4] ✉ & Yihao Yang [1,2,3,4] ✉

Higher-order Weyl semimetals are a family of recently predicted topological phases simultaneously showcasing unconventional properties derived from Weyl points, such as chiral anomaly, and multidimensional topological phenomena originating from higher-order topology. The higher-order Weyl semimetal phases, with their higher-order topology arising from quantized dipole or quadrupole bulk polarizations, have been demonstrated in phononics and circuits. Here, we experimentally discover a class of higher-order Weyl semimetal phase in a three-dimensional photonic crystal (PhC), exhibiting the concurrence of the surface and hinge Fermi arcs from the nonzero Chern number and the nontrivial generalized real Chern number, respectively, coined a real higher-order Weyl PhC. Notably, the projected two-dimensional subsystem with $k_z = 0$ is a real Chern insulator, belonging to the Stiefel-Whitney class with real Bloch wavefunctions, which is distinguished fundamentally from the Chern class with complex Bloch wavefunctions. Our work offers an ideal photonic platform for exploring potential applications and material properties associated with the higher-order Weyl points and the Stiefel-Whitney class of topological phases.

Weyl semimetals and their classical analogues feature two-fold linear band crossings in three-dimensional (3D) momentum space, known as Weyl points[1–5], analogous to the Dirac points in two-dimensional (2D) momentum space. These Weyl points act as monopoles of Berry flux and carry topological chiral charges defined by the Chern number. Consequently, the 2D Fermi-arc surface states are formed, connecting the projections of two oppositely charged Weyl points, according to the celebrated bulk-boundary correspondence. On the other hand, the recent discovery of higher-order topological phases has revolutionized the study of topological matters[6–16]. The higher-order topological phases give rise to unconventional bulk-boundary correspondence as they host boundary states in at least two dimensions lower than the bulk, in contrast to the previous first-order topological phases, where

the topological boundary states living in just one dimension lower than the bulk.

Higher-order topology can also be incorporated into Weyl semimetals, resulting in higher-order Weyl semimetals that possess simultaneously chiral Fermi-arc surface states in two dimensions and Fermi-arc hinge states in one dimension[17–22]. These states stem from the chiral charge and the higher-order charge of Weyl points, respectively, and bridge the projections of Weyl points in multiple dimensions, thereby revealing the dimensional hierarchy of higher-order topological physics. In sharp contrast to the extensive study of higher-order topology in insulators, their Weyl semimetal counterparts have been severely lagged behind, with only a handful of experimental demonstrations in phononics[19,20,22] and circuits[21], where the couplings can be engineered

[1]Interdisciplinary Center for Quantum Information, State Key Laboratory of Extreme Photonics and Instrumentation, ZJU-Hangzhou Global Scientific and Technological Innovation Center, Zhejiang University, Hangzhou 310027, China. [2]International Joint Innovation Center, The Electromagnetics Academy at Zhejiang University, Zhejiang University, Haining 314400, China. [3]Key Lab. of Advanced Micro/Nano Electronic Devices & Smart Systems of Zhejiang, Jinhua Institute of Zhejiang University, Zhejiang University, Jinhua 321099, China. [4]Shaoxing Institute of Zhejiang University, Zhejiang University, Shaoxing 312000, China. [5]Centre for Quantum Physics, Key Laboratory of Advanced Optoelectronic Quantum Architecture and Measurement (MOE), School of Physics, Beijing Institute of Technology, Beijing 100081, China. [6]Beijing Key Laboratory of Nanophotonics and Ultrafine Optoelectronic Systems, School of Physics, Beijing Institute of Technology, Beijing 100081, China. [7]These authors contributed equally: Yuang Pan, Chaoxi Cui, Qiaolu Chen. ✉e-mail: zhiming_yu@bit.edu.cn; hansomchen@zju.edu.cn; yangyihao@zju.edu.cn

flexibly to implement discrete lattice models. Nevertheless, the higher-order Weyl semimetal phases in photonics remain uncharted territory, both theoretically and experimentally, due to the inherent challenge of discretely modeling most photonic systems, particularly those in three dimensions.

Our work reports on the experimental discovery of a higher-order Weyl photonic crystal (PhC). It differs from the previously achieved higher-order topological Weyl semimetal phases[19–22] in following three ways. First, the newly proposed higher-order Weyl points separate the 3D Brillouin zone (BZ) into the Chern insulator phase and the generalized real Chern insulator phase[23,24], as opposed to the previously discovered higher-order Weyl points that separate the BZ into the Chern insulator phase and a higher-order topological phase with quadrupole or dipole bulk polarizations[19–22]. Particularly, the $k_z = 0$ slice can be viewed as a previously overlooked real Chern insulator, belonging to the Stiefel-Whitney class with real Bloch wavefunctions, which is fundamentally different from the Chern class with complex Bloch wavefunctions[25]. We thus coin the proposed Weyl points "real higher-order Weyl points". Second, the real higher-order Weyl points are achieved through a symmetry-enforced topological phase transition process, where a three-fold spin-1 Weyl point[26] splits into two real higher-order Weyl points. Though inspired by a tight-binding model, our design principle is symmetry-guided, beyond the conventional lattice models that unfavorable to many physical systems, as exemplified by the PhCs (see Supplementary Information Note 3 and Fig. S4). Third, our work provides the first example of higher-order photonic Weyl points, manifesting the dimensional hierarchy of higher-order topological physics in a 3D PhC. All previous photonic Weyl points were limited to the first-order[27–33]. Interestingly, our Weyl points are isolated from other trivial bands, representing a higher-order version of the ideal Weyl points[31,33,34]. The realization of ideal higher-order photonic Weyl points paves a way toward the robust manipulation of the flow of light across multiple dimensions, including 2D surfaces and 1D hinges, in a single integrable 3D platform.

## Results

### Design of the real higher-order Weyl PhC

The 3D cubic unit cell of the PhC with lattice constant $a = 15$ mm is displayed in Fig. 1a. Each unit cell contains four junctions, at $(x, y, z)$, $(-x + 0.5a, -y, z + 0.5a)$, $(-x, y + 0.5a, -z + 0.5a)$, $(x + 0.5a, -y + 0.5a, -z)$, respectively, where $x = y = 0.3a$, $z = 0.2a$. Each junction connects to four neighboring junctions via square perfect electric conductor (PEC) rods with two different widths $w_1 = 3.5$ mm (green rods), $w_2 = 4.5$ mm (red rods); the rest of the volume is filled with free space. The resulting 3D PhC has the non-centrosymmetric space group $P2_12_12_1$ (No. 19) with three two-fold screw symmetries, $S_{2x} := (x, y, z) \rightarrow (x + 0.5a, -y + 0.5a, -z)$, $S_{2y} := (x, y, z) \rightarrow (-x, y + 0.5a, -z + 0.5a)$, and $S_{2z} := (x, y, z) \rightarrow (-x + 0.5a, -y, z + 0.5a)$.

The band structure of the 3D PhC has been numerically calculated, as depicted in Fig. 1b, c, d, e, g. The analysis reveals two ideal real higher-order two-fold Weyl points at $k = (0, 0, \pm k_{wp})$ (illustrated as yellow dots), and an ideal charge-2 four-fold 3D Dirac point at R (indicated by red dots), which are isolated from the other trivial bands[31,33]. Note that, the pair of Weyl points can move along the $k_z$ axis, by tuning the geometric parameters but preserving the lattice symmetry; for the convenience of measurement, $k_{wp} = 0.5 \, \pi/a$ in our experiments. Besides, the resulting 3D Dirac point carries a chiral charge $+2$[26], which is a direct sum of two identical Weyl points, in stark contrast to the conventional 3D Dirac points, which is a sum of two oppositely charged Weyl points[35,36]. Interestingly, there exist three nodal surfaces on the $k_x = \pi/a$, $k_y = \pi/a$, and $k_z = \pi/a$ planes, carrying a vanishing chiral charge, in contrast to the previously demonstrated nodal surface with a chiral charge 2[37,38].

The two real higher-order Weyl points divide the 3D BZ into three sections parametrized by $k_z$. As depicted in Fig. 1f, each section can be viewed as a $k_z$-dependent 2D phase characterized by the Chern number

$$C_n = \frac{1}{2\pi} \int_{BZ} \Omega_{n,xy} d^2 k \qquad (1)$$

where $\Omega_{n,xy}$ is Berry curvature and $k$ is the wavevector. Via the first-principle calculation as well as the symmetry indicator analysis shown in Fig. 1g, we can prove that the $|k_z| < k_{wp}$ planes possess a vanishing Chern number 0, while the $k_z > k_{wp}$ ($k_z < -k_{wp}$) planes have a Chern number $+1$ ($-1$) (see Supplementary Information Note 8 and Fig. S9). The higher-order Weyl points as the transition point, thus, carry a chiral charge $+1$. Consequently, within the $|k_z| > k_{wp}$ region, the chiral edge states emerge in the 2D subsystems, corresponding to the chiral surface Fermi arcs connecting the projections of the higher-order Weyl points and the Dirac point in the 3D system.

Though the 2D subsystems between two Weyl points are topologically trivial in terms of the Chern number, it is topologically nontrivial in terms of the generalized real Chern number ($v_R$) that characterizes the higher-order topology. This generalized real Chern number is well defined in the $C_{2z}$-invariant system with a vanishing Chern number and calculated from the $C_{2z}$ eigenvalues at $C_{2z}$-invariant momenta

$$(-1)^{v_R} = \prod_{i=1}^{4} (-1)^{[N_{occ}^-(\Gamma_i)/2]} \qquad (2)$$

where $\Gamma_i$ denotes the four $C_{2z}$-invariant momenta at $\Gamma$, X, Y and S, and $N_{occ}^-(\Gamma_i)$ is the number of occupied bands with negative inversion eigenvalues at $\Gamma_i$, with the bracket denoting the greatest integer function. The $C_{2z}$ eigenvalues are calculated from the TB models (see Supplementary Information Note 1 and Table. S1). The $C_{2z}$ eigenvalues of the lowest four bands at $\Gamma$, X, Y and S are displayed in Fig. 1g, from which a nontrivial $v_R$ can be obtained for the $|k_z| < k_{wp}$ plane. Particularly, the $k_z = 0$ slice has the $C_{2z}T$ symmetry (the combination of two-fold rotation and time-reversal symmetry) that enforces the bands to be real[23–25], as opposite to the complex bands in the previous higher-order topological insulator phases[6,7,11,13–16]. Such a higher-order topological phase is known as the real Chern insulator[23,24], belonging to the previously overlooked Stiefel-Whitney class[25]. Though at the $k_z \neq 0$ planes, the bands are not real, the higher-order topology is still determined by the generalized real Chern number, we thus term all the $|k_z| < k_{wp}$ planes as the generalized real Chern insulator (see Supplementary Information Note 8 and Fig. S9). The nontrivial generalized real Cheren number in the $|k_z| < k_{wp}$ region gives rise to the corner states in the 2D subsystems, corresponding to the 1D hinge Fermi arcs protected by the $C_{2z}$ symmetry in the 3D system. Interestingly, the $|k_z| > k_{wp}$ planes also host first-order topological index, i.e., nontrivial Zak phase, resulting in the floating surface states at the (100) and (010) surfaces (see Supplementary Information Note 9 and Fig. S10).

Note that the current structure is transformed from the PhC with the space group No. 198 that has three $C_2$ screw symmetry along the $x$, $y$, $z$ axis, and a $C_3$ rotational symmetry along the <111> axis. The original PhC hosts the symmetry-enforced spin-1 Weyl point and the charge-2 Dirac point. Interestingly, the $C_{3,III}$ symmetry is crucial for the spin-1 Weyl point, but it is not necessary for the existence of the charge-2 Dirac point. By breaking the $C_{3,III}$ symmetry, the spin-1 Weyl point splits into two real higher-order Weyl points, while the charge-2 Dirac point persists; the corresponding space group is reduced to No. 19 (see Supplementary Information Note 3 and Fig. S4).

### Experimental demonstration of the real higher-order Weyl points

As depicted in Fig. 2a, b, the 3D metallic PhC can be fabricated directly from AlSi$_{10}$Mg (acting approximately as PEC at microwave frequencies,

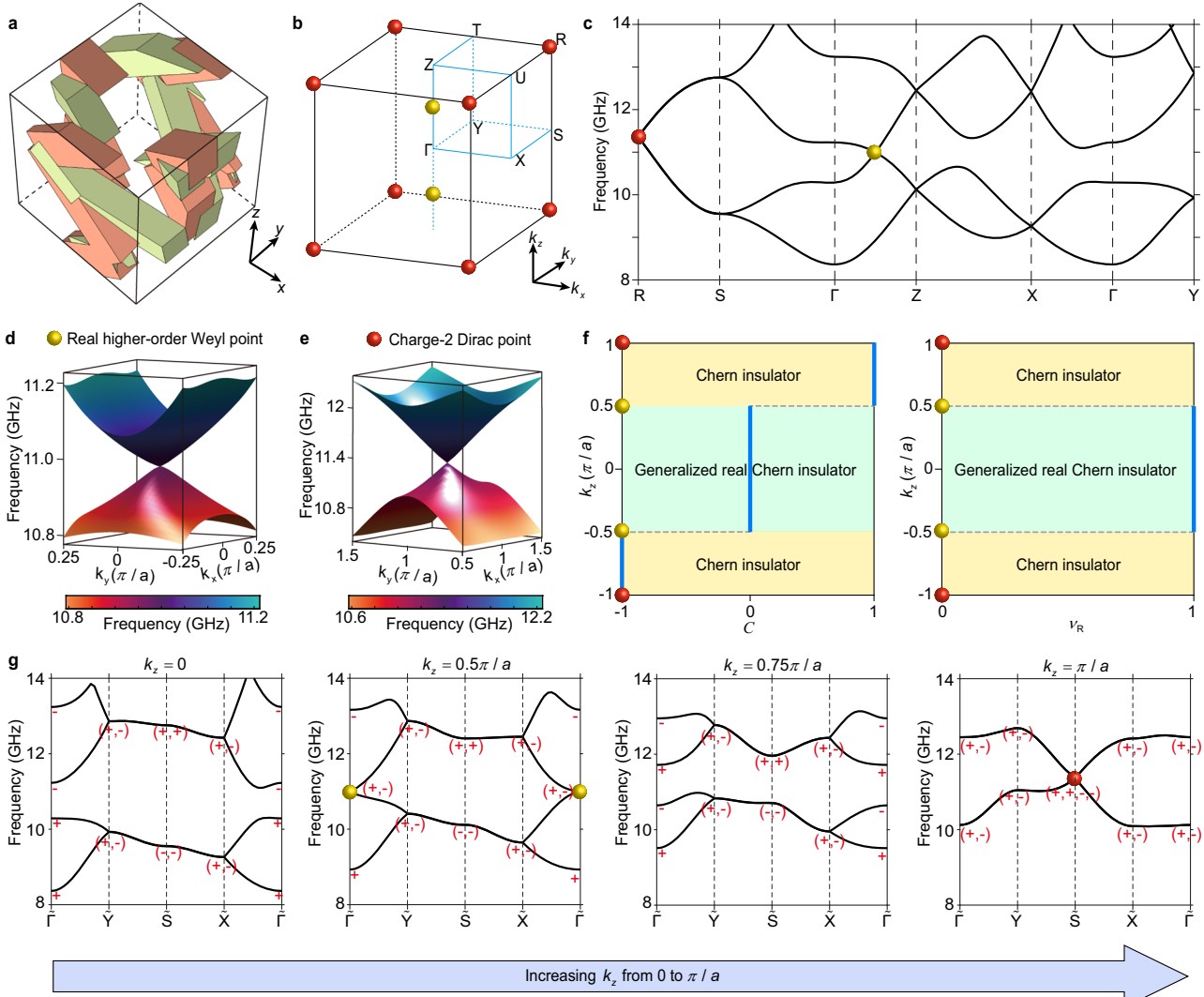

**Fig. 1 | 3D real higher-order Weyl PhC. a** Unit cell of the 3D PhC. The red and green regions represent square perfect electric conductor (PEC) rods with width $w_1 = 4.5$ mm and $w_2 = 3.5$ mm, respectively. **b** 3D Brillouin zone of the PhC. **c** 3D Band structure of the PhC. **d** 2D band structure in the vicinity of the real higher-order Weyl point. **e** 2D band structure in the vicinity of the charge-2 Dirac point. **f** Chern number $C$ and generalized real Chern number $v_R$ in the projected 2D subsystems parameterized by $k_z$. **g** Band structures of the 2D projected subsystems with $k_z = 0$, $0.5 \pi/a$, $0.75 \pi/a$, and $\pi/a$, respectively. The $C_{2z}$ eigenvalues at $\Gamma$, Y, S and X of the four lowest bands are marked. The positive and negative sign represents the $C_{2z}$ eigenvalue of +1 and −1, respectively.

see Supplementary Information Note 10 and Fig. S11) via additive manufacturing techniques, owing to its self-supporting structure. The holes on the vertical surfaces of the sample allow us to insert the probe into the sample to measure the field distributions. To determine the bulk band structure, we conduct microwave pump-probe measurements on the sample. As illustrated in Fig. 2c, the source is vertically inserted into the middle of the sample to excite the bulk modes, and the probe is horizontally inserted into the sample to measure the complex field distribution along a vertical plane situated 15 mm away from the source. After performing 2D Fourier transform to the measured real-space field distribution, we obtain the bulk band structure projected onto the $k_y$-$k_z$ plane, as indicated in Fig. 2d. The results, presented in Fig. 2e, reveal that the real higher-order Weyl point is projected onto the middle of the $\bar{\Gamma} - \bar{Z}$ path at 10.9 GHz, and the charge-2 Dirac point is projected onto $\bar{M}$ at 11.3 GHz. The measurement results are in good agreement with the simulated projected bulk dispersion shown in Fig. 2f.

Next, we perform experiments to measure the topological surface Fermi arcs from the real higher-order Weyl points on the (100) surface. The experimental setup configuration is depicted in

Fig. 3a. All vertical surfaces of the sample are covered with a 0.75 mm thick metallic layer, acting as PEC boundaries. A source is positioned at the center of the surface to excite the surface modes. Upon Fourier-transforming the measured field distributions from real space to reciprocal space, we acquire the surface dispersion depicted in Fig. 3b, c, e. The insert in Fig. 3b displays the momentum space distribution corresponding to the filed pattern shown in Fig. 3a. The two open arcs are the photonic Fermi arcs connecting the projections of the charge-2 3D Dirac point and the real higher-order Weyl points. The measured surface dispersion along the high symmetry line $\bar{\Gamma} - \bar{Z} - \bar{M} - \bar{Y} - \bar{\Gamma}$ is shown in Fig. 3c, demonstrating excellent consistence with the simulated surface dispersion (red lines) depicted in Fig. 3d. In addition, we present the measured two-dimensional surface isofrequency contours at frequencies ranging from 10.9 to 11.9 GHz in Fig. 3e, corroborating the outstanding concurrence with the simulated outcomes in Fig. 3f. Apart from the Fermi arc surface states, the topological floating surface states protected by nontrivial Zak phase in the $|k_z| < k_{wp}$ plane also exist on the surface, as shown in Fig. 3c, d, along $\bar{Y} - \bar{\Gamma}$ and $\bar{\Gamma} - \bar{Z}$. These floating surface states coexist with the Fermi arc surface states.

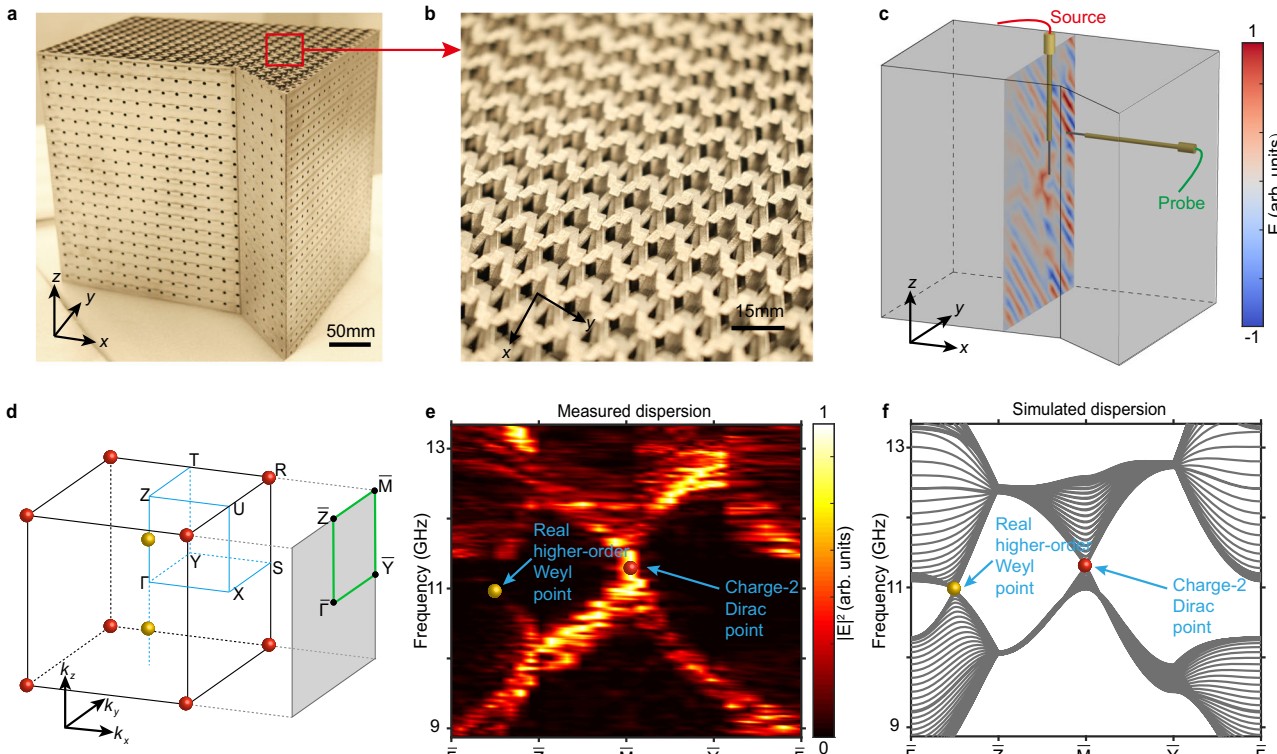

**Fig. 2 | Projected bulk dispersion of the real higher-order Weyl points.**
**a, b** Photographs of the fabricated 3D PhC. **c** Experimental setup. The field distributions are measured along the vertical plane in the middle of the sample. **d** Schematic view of the projected 2D surface BZ. **e** Measured projected bulk dispersion. The color bar indicates the energy intensity. **f** Simulated projected bulk dispersion.

Finally, we experimentally characterize the topological hinge Fermi arc originating from the real higher-order Weyl points. A hinge between (100) and (110) surfaces is selected, by analyzing the Wannier centers (see Supplementary Information Note 1 and Fig. S3). To excite the hinge modes, a source is placed at the middle of the hinge. Subsequently, we employ a probe to scan the field distributions along the hinge. The experimental setup and the measured field distributions on the two surfaces adjacent to the hinge is shown in Fig. 4a. The field is predominantly localized on the sample's hinge, signifying the presence of topological hinge states. The measured hinge dispersion along the $k_z$ direction is obtained through Fourier transformation, as presented in Fig. 4b, which shows excellent agreement with the simulated predictions depicted in Fig. 4d (red line). To thoroughly and clearly illustrate the characteristics of our sample, in which the real higher-order Weyl points separate the BZ zone into the generalized real Chern insulator and the Chern insulator in the $k_z$ direction, we further plot the measured energy intensity for varying $k_z$ in a small area around the hinge of the sample, as shown in Fig. 4c. By exciting both the hinge and surface modes, it is evident that the hinge Fermi arcs manifest themselves well for $|k_z| < 0.5\,\pi/a$, connecting the projection of the two Weyl points.

## Discussion

We have thus accomplished successfully the experimental realization of an ideal real higher-order Weyl PhC exhibiting simultaneously one-dimensional (1D) hinge Fermi arcs, 2D surface Fermi arcs, and 2D floating surface states, originating from the nontrivial generalized real Chern number, Chern number, and Zak phase, respectively. The distinct topological origins of the boundary states pave a way toward the robust manipulation of the flow of light and creation of photonic devices across multiple dimensions, including the 2D surfaces and the 1D hinges, in a single integrable 3D PhC. Besides, our work provides the experimental evidences of the

charge-2 3D Dirac point and the 2D nodal surface with the vanishing chiral charge, neither of which has been realized previously in photonics. Finally, our findings broaden our understanding of the higher-order Weyl semimetals and the Stiefel-Whitney class of topological phases, and establish an ideal photonic platform to explore exotic physical phenomena related to higher-order Weyl points, such as chiral anomaly, pseudo-gauge fields, and fractional charges.

## Methods

### Numerical simulations

All simulations are performed using the COMSOL Multiphysics software package. The metallic material of the 3D PhC is considered as PEC, and the rest volume is air. To calculate the bulk dispersion of the unit cell, periodic boundary conditions are applied in all three spatial directions. To calculate the surface dispersion, we consider a super cell composed of 15 unit cells; Periodic boundary conditions are imposed in the $y$ and $z$ directions, and PEC boundary conditions in the $x$ direction. For the hinge dispersion, the supercell has 11 by 11 unit cells; Periodic boundary conditions are imposed in the $z$ direction, and PEC boundary conditions in the $x$ and $y$ directions. In simulation, the size of the system has been varied to confirm convergence (see Supplementary Information Note 11, Figs. S12 and S13).

### Experiment

The sample is fabricated via 3D metal printing. The material is $AlSi_{10}Mg$, with high conductivity at microwave frequencies. Notably, AlSi10Mg at microwave frequencies has a negligible metallic loss, which can be regarded approximately as PEC (see Supplementary Information Note 10 and Fig. S11). In the measurements, the amplitude and phase of the fields are collected by a vector network analyzer (VNA). The VNA is connected to two electric dipole antennas, serving as the source and the probe, respectively.

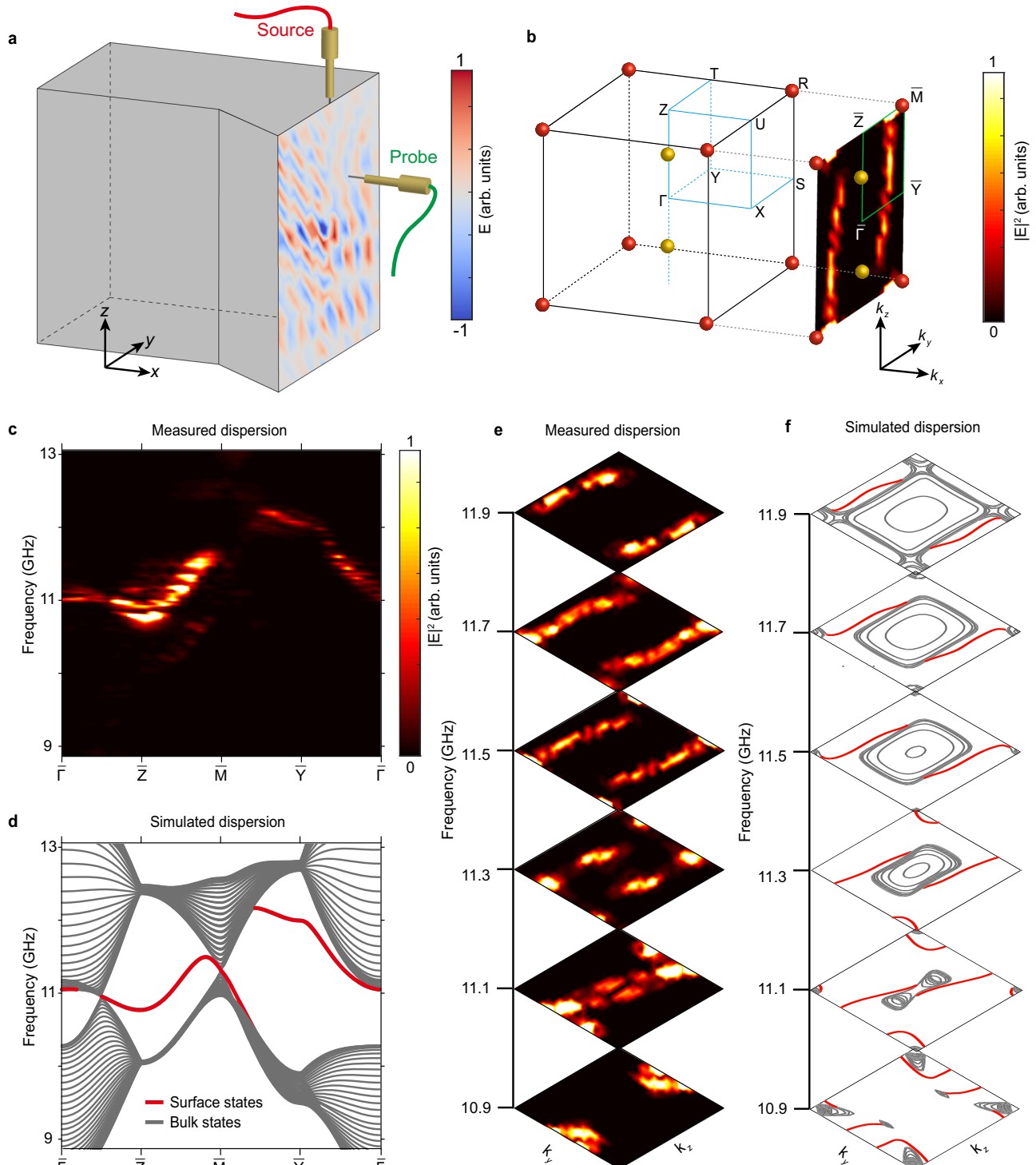

**Fig. 3 | Surface Fermi arcs from the real higher-order Weyl points.**
**a** Experimental setup. The field distributions are measured on the (100) sample surface covered by a metallic layer. **b** Schematic of the 2D surface BZ. The right inset shows measured results obtained by Fourier transforming the field pattern measured in (**a**). **c** Measured surface dispersions along high-symmetry lines. **d** Simulated surface dispersions along high-symmetry lines. **e** Measured surface isofrequency contours from 10.9 to 11.9 GHz. The color map measures the energy intensity. **f** Simulated surface isofrequency contours from 10.9 to 11.9 GHz.

To excite the bulk states, the source is vertically inserted into the middle of the sample, and the probe is horizontally inserted into the sample to measure the field distribution along a vertical plane 15 mm away from the source. To excite the surface states, A source is placed at the center of the surface; the distance between the source and the measured plane is about 11 mm. For the hinge states, the source is placed at the middle of the hinge, and the probe is moved along the hinge to scan the field.

## Data availability
The data that support the findings of this study are available at https://doi.org/10.5281/zenodo.8375373.

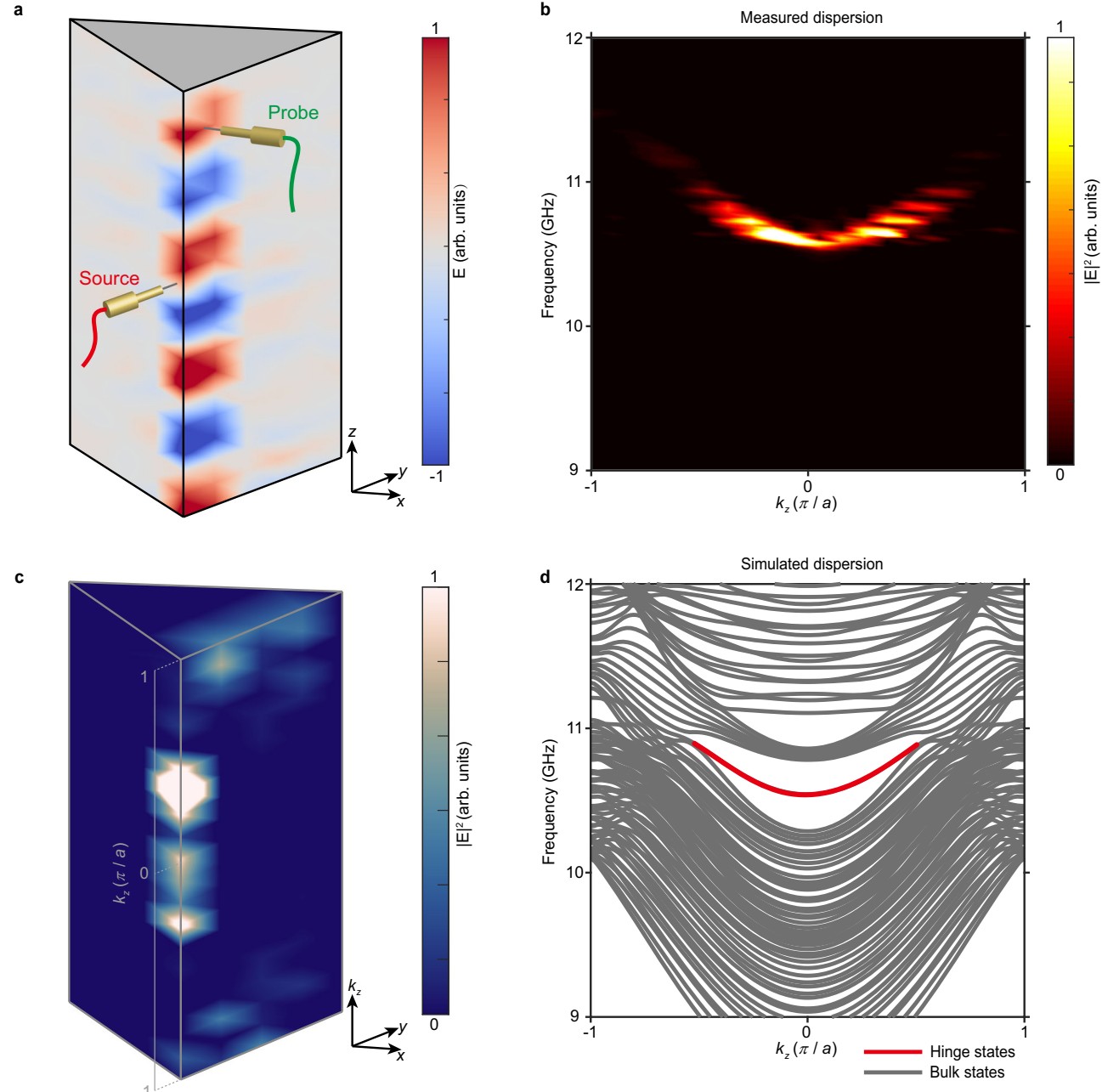

**Fig. 4 | Hinge Fermi arc from the real higher-order Weyl points. a** Measured field distributions of the two surfaces adjacent to the hinge at 10.7 GHz. **b** Measured projected hinge dispersion along the $k_z$ direction. **c** Measured energy intensity for varying $k_z$ in a small area around the hinge at 10.7 GHz, with 5, 5, and 20 unit cells in the $x$, $y$, and $z$ direction respectively. **d** Simulated projected hinge dispersion along the $k_z$ direction.

## Code availability

The codes that support the findings of this study are available at https://doi.org/10.5281/zenodo.8375373.

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

## Acknowledgements

The work at Zhejiang University was sponsored by the Key Research and Development Program of the Ministry of Science and Technology under Grants 2022YFA1405200 (Y.Y.), No. 2022YFA1404704 (H.C.), 2022YFA1404902 (H.C.), and 2022YFA1404900 (Y.Y.), the National Natural Science Foundation of China (NNSFC) under Grants No. 62175215 (Y.Y.), No. 61975176 (H.C.), and No. 12004035 (Z.Y.), the Key Research and Development Program of Zhejiang Province under Grant No.2022C01036 (H.C.), the Fundamental Research Funds for the Central Universities (2021FZZX001-19) (Y.Y.), and the Excellent Young Scientists Fund Program (Overseas) of China (Y.Y.)

## Author contributions

Y.Y. and Z.Y. initiated the idea. Y.Y. and Y.P. designed the experiment, Y.P. and Y.Y. fabricated samples. Y.P. carried out the measurement with the assistance from Q.C., F.C., and Y.R. Y.P. analyzed the data. Y.P., Y.Y., and Q.C. performed the simulations. Z.Y., C.C., Y.Y., Y.P., L.Z., N.H., W.L., and X.L. did the theoretical analysis. Y.P., Y.Y., Z.Y., and C.C. wrote the paper. Y.Y., H.C., and Z.Y. supervised the project. All authors participated in discussions and reviewed the paper.

## Competing interests

The authors declare no competing interests.
