## [Peer Review File · Nature Communications]

REVIEWER COMMENTS

Reviewer #1 (Remarks to the Author):

In the manuscript entitled “Real higher-order Weyl photonic crystal”, the authors report the first experimental realization of the photonic higher-order Weyl semimetal. Through an ingenious symmetry-enforced topological transition process, they create two real higher-order Weyl points from a three-fold spin-1 Weyl point. Interestingly, the higher-order Weyl points in this work are isolated from other trivial bands, representing a higher-order version of the ideal Weyl points. They find that the real higher-order Weyl points separate the 3D Brillouin zone (BZ) into the Chern insulator phase and the generalized real Chern insulator phase, as opposed to the previously discovered higher-order Weyl points that separate the BZ into the Chern insulator phase and a higher-order topological phase with quadrupole or dipole bulk polarizations. Notably, the authors prove that the $k_z = 0$ slice can be regarded as a real Chern insulator, belonging to the Stiefel-Whitney class with real Bloch wavefunctions. Using the additive manufacturing techniques and microwave pump-probe measurements, they experimentally observe the 1D hinge Fermi arcs and 2D surface Fermi arcs in a single 3D photonic crystal. The experimental observations agree with the simulated results.

This work signifies an important research advancement in photonic higher-order Weyl semimetals and Stiefel-Whitney topological phases, which is captivating and timely. The symmetry-guided design principle proposed in this work has great guiding significance for designing higher-order Weyl semimetal phases beyond the conventional lattice models. The experimental results are truly remarkable, offering compelling evidence, while the theoretical analysis is solid and reliable. In addition, the manuscript is well written and organized, and is easy to follow. In my opinion, this work reaches the high criteria of Nature Communications, and I am happy to recommend the publication in Nature Communications.

Here, I place several suggestions and comments for authors' consideration:

1. In Fig. 1g, are the C_{2z} eigenvalues at the high-symmetry points calculated from the first-principle simulations or the TB models? The authors should explain the details.
2. In Fig. 2a, the authors present the photograph of the fabricated sample and there are holes on the vertical surfaces of the sample. What is the purpose of these holes?
3. Fig. S6 only lists the instruments and samples used in the measurement. It would be better for the authors to present the experimental setups of the bulk, surface and hinge measurements in three different figures, respectively.

4. In Fig. S2, the authors only present the band structures of the TB models. I suggest the authors to supplement the corresponding 3D Brillouin zone.

Reviewer #2 (Remarks to the Author):

The study reports the experimental discovery of a higher-order Weyl photonic crystal (PhC) in a three-dimensional system. The discovered higher-order Weyl points in the PhC separate the three-dimensional Brillouin zone into the Chern insulator phase and the generalized real Chern insulator phase. These points are referred to as "real higher-order Weyl points" and exhibit properties that distinguish them from previously known higher-order Weyl points. The PhC exhibits the concurrence of surface Fermi arcs and hinge Fermi arcs. Surface Fermi arcs are derived from the nonzero Chern number, while hinge Fermi arcs result from the nontrivial generalized real Chern number. The presence of both types of Fermi arcs highlights the unique characteristics of the higher-order Weyl semimetal phase in the PhC. The projected two-dimensional subsystem of the PhC, specifically the $k_z = 0$ slice, is identified as a real Chern insulator. This insulating phase belongs to the Stiefel-Whitney class with real Bloch wavefunctions, which distinguishes it from the conventional Chern class with complex Bloch wavefunctions.

The study explores the relatively uncharted territory of higher-order Weyl semimetals in photonics. While higher-order topological phases have been extensively studied in insulators, their counterparts in Weyl semimetals have received less attention. By experimentally discovering and characterizing a higher-order Weyl PhC, the study contributes to expanding the understanding of higher-order topological phenomena in Weyl semimetals, which has implications for the broader field of topological materials. The identification and characterization of "real higher-order Weyl points" in the PhC introduce a new class of higher-order Weyl semimetals. This distinction sets them apart from previously known higher-order Weyl points that exhibited different characteristics. The existence of real higher-order Weyl points and their associated surface and hinge Fermi arcs highlight novel features of the PhC system and contribute to the growing knowledge of topological materials. The recognition of the $k_z = 0$ slice of the PhC as a real Chern insulator belonging to the Stiefel-Whitney class provides insights into the unique topological properties of the system. This finding expands the understanding of different classes of topological insulators and their associated Bloch wavefunctions. It also offers new possibilities for designing and manipulating photonic systems based on the Stiefel-Whitney class. Therefore, this work is of interest to a broad audience, and I recommend its publication in Nat. Common. I have only a few minor comments that should be addressed before publication.

1. How is AlSi10Mg compared to more conventional metals used in experiments like Au or Ag? Have they also been used as PEC in the experiment for comparison?

2. For COMSOL simulation, has the size of the system been varied to confirm convergence?

3. A few minor issues. Headings can be helpful in the main article to help readers follow the logic. It is also helpful to special the sections when referring to the supplementary information. Also, there are a few typos including integratable (integrable) and tirival (trivial).

Response Letter to Reviewers

GENERAL COMMENTS FROM 1st REVIEWER:

In the manuscript entitled “Real higher-order Weyl photonic crystal”, the authors report the first experimental realization of the photonic higher-order Weyl semimetal. Through an ingenious symmetry-enforced topological transition process, they create two real higher-order Weyl points from a three-fold spin-1 Weyl point. Interestingly, the higher-order Weyl points in this work are isolated from other trivial bands, representing a higher-order version of the ideal Weyl points. They find that the real higher-order Weyl points separate the 3D Brillouin zone (BZ) into the Chern insulator phase and the generalized real Chern insulator phase, as opposed to the previously discovered higher-order Weyl points that separate the BZ into the Chern insulator phase and a higher-order topological phase with quadrupole or dipole bulk polarizations. Notably, the authors prove that the $k_z = 0$ slice can be regarded as a real Chern insulator, belonging to the Stiefel-Whitney class with real Bloch wavefunctions. Using the additive manufacturing techniques and microwave pump-probe measurements, they experimentally observe the 1D hinge Fermi arcs and 2D surface Fermi arcs in a single 3D photonic crystal. The experimental observations agree with the simulated results.

This work signifies an important research advancement in photonic higher-order Weyl semimetals and Stiefel-Whitney topological phases, which is captivating and timely. The symmetry-guided design principle proposed in this work has great guiding significance for designing higher-order Weyl semimetal phases beyond the conventional lattice models. The experimental results are truly remarkable, offering compelling evidence, while the theoretical analysis is solid and reliable. In addition, the manuscript is well written and organized, and is easily to follow. In my opinion, this work reaches the high criteria of Nature Communications, and I am happy to recommend the publication in Nature Communications.

Response from Authors:

We thank the reviewer for the positive comments and the favorable recommendation for publication of this work in *Nature Communications*. In the following, we will fully address the specific comments point-by-point.

SPECIFIC COMMENTS FROM 1st REVIEWER:

1st Reviewer -- Comment 1:

In Fig. 1g, are the C_{2z} eigenvalues at the high-symmetry points calculated from the first-principle simulations or the TB models? The authors should explain the details.

Response from Authors:

We thank the reviewer for the comments.

The C_{2z} eigenvalues are calculated from the tight-binding (TB) models. The C_{2z} eigenvalues of a state is calculated as follow. At a C_{2z} invariant point K , C_{2z} eigenvalues of n^{th} band is defined as

$$\zeta_n = \langle \psi_{nK} | \hat{C}_{2z} | \psi_{nK} \rangle, \quad (1)$$

where \hat{C}_{2z} is the C_{2z} operator and $|\psi_{nK}\rangle$ is the wave function of n^{th} band at K . For the TB model,

the Hamiltonian and wave functions are written in a basis $\{|\phi_{\alpha K}\rangle\}$,

$$|\phi_{\alpha K}\rangle = \frac{1}{N} \sum_{i=1}^N e^{i\mathbf{K}\cdot(\mathbf{R}_i + \mathbf{r}_\alpha)} |\phi_{\alpha R_i}\rangle, \quad (2)$$

where $|\phi_{\alpha R_i}\rangle$ is the α^{th} orbit in R_i cell. In this basis, wave function can be expanded as

$$|\psi_{nK}\rangle = \sum_{\alpha=1}^{N_{orbit}} c_{n\alpha} |\phi_{\alpha K}\rangle, \quad (3)$$

$c_{n\alpha}$ is the α^{th} element of n^{th} eigenvector of Hamiltonian $H(k)$. By combining Eq. (1) and (3), one can get the expression of C_{2z} eigenvalue of n^{th} band as

$$\zeta_n = \sum_{\alpha=1}^{N_{orbit}} \sum_{\beta=1}^{N_{orbit}} c_{n\beta}^* C_{2z\beta\alpha} e^{2i\mathbf{K}\cdot\mathbf{r}_\alpha} c_{n\alpha}, \quad (4)$$

where $C_{2z\beta\alpha}$ is the C_{2z} matrix under the basis of $\{|\phi_\alpha\rangle\}$. This can be rewritten into a compact matrix form

$$\zeta_n = c_n^\dagger C_{\beta\alpha} D_K c_n, \quad (5)$$

where $D_K = \text{Diag}(e^{2iKr_1}, e^{2iKr_2}, \dots)$.

Accordingly, we have added the above details in Supplementary Note 1.

We have also added a discussion referring to Supplementary Note 1, on line 136, page 4 of the main text, which reads as

“The C_{2z} eigenvalues are calculated from the TB models (see Supplementary Information Note 1).”

1st Reviewer -- Comment 2:

In Fig. 2a, the authors present the photograph of the fabricated sample and there are holes on the vertical surfaces of the sample. What is the purpose of these holes?

Response from Authors:

The purpose of these holes is to allow us to insert the probe into the sample to measure the field distributions.

Accordingly, we have added a sentence on line 172, page 6 of the main text, which reads as,

“The holes on the vertical surfaces of the sample allow us to insert the probe into the sample to measure the field distributions”.

1st Reviewer -- Comment 3:

Fig. S6 only lists the instruments and samples used in the measurement. It would be better for the authors to present the experimental setups of the bulk, surface, and hinge measurements in three different figures, respectively.

Response from Authors:

We thank the reviewer for this constructive suggestion.

Following the reviewer's suggestion, we have added experimental setups of the bulk, surface, and hinge measurements in Fig. S6 in Supplementary Note 5.

Fig. S6 Experimental setups for measuring bulk (a), surface (b), and hinge (c) states.

1st Reviewer -- Comment 4:

In Fig. S2, the authors only present the band structures of the TB models. I suggest the authors to supplement the corresponding 3D Brillouin zone.

Response from Authors:

We thank the reviewer for this good suggestion.

Following the reviewer's suggestion, we have added the 3D Brillouin zone in Fig. S2 in Supplementary Note1.

Fig. S2 a, Band structure of TB model (1) in space group No.198. b, Perturbed band structure. The inserts show the corresponding 3D Brillouin zones.

GENERAL COMMENTS FROM 2nd REVIEWER:

The study reports the experimental discovery of a higher-order Weyl photonic crystal (PhC) in a three-dimensional system. The discovered higher-order Weyl points in the PhC separate the three-dimensional Brillouin zone into the Chern insulator phase and the generalized real Chern insulator phase. These points are referred to as "real higher-order Weyl points" and exhibit properties that distinguish them from previously known higher-order Weyl points. The PhC exhibits the concurrence of surface Fermi arcs and hinge Fermi arcs. Surface Fermi arcs are derived from the nonzero Chern number, while hinge Fermi arcs result from the nontrivial generalized real Chern number. The presence of both types of Fermi arcs highlights the unique characteristics of the higher-order Weyl semimetal phase in the PhC. The projected two-dimensional subsystem of the PhC, specifically the $k_z = 0$ slice, is identified as a real Chern insulator. This insulating phase belongs to the Stiefel-Whitney class with real Bloch wavefunctions, which distinguishes it from the conventional Chern class with complex Bloch wavefunctions.

The study explores the relatively uncharted territory of higher-order Weyl semimetals in photonics. While higher-order topological phases have been extensively studied in insulators, their counterparts in Weyl semimetals have received less attention. By experimentally discovering and characterizing a higher-order Weyl PhC, the study contributes to expanding the understanding of higher-order topological phenomena in Weyl semimetals, which has implications for the broader field of topological materials. The identification and characterization of "real higher-order Weyl points" in the PhC introduce a new class of higher-order Weyl semimetals. This distinction sets them apart from previously known higher-order Weyl points that exhibited different characteristics. The existence of real higher-order Weyl points and their associated surface and hinge Fermi arcs highlight novel features of the PhC system and contribute to the growing knowledge of topological materials. The recognition of the $k_z = 0$ slice of the PhC as a real Chern insulator belonging to the Stiefel-Whitney class provides insights into the unique topological properties of the system. This finding expands the understanding of different classes of topological insulators and their associated Bloch wavefunctions. It also offers new possibilities for designing and manipulating photonic systems based on the Stiefel-Whitney class. Therefore, this work is of interest to a broad audience,

and I recommend its publication in Nat. Common. I have only a few minor comments that should be addressed before publication.

Response from Authors:

We thank the reviewer for the positive comments and the favorable recommendation for publication of this work in *Nature Communications*. In the following, we will fully address the specific comments point-by-point.

SPECIFIC COMMENTS FROM 1st REVIEWER:

2nd Reviewer -- Comment 1:

How is AlSi10Mg compared to more conventional metals used in experiments like Au or Ag? Have they also been used as PEC in the experiment for comparison?

Response from Authors:

We thank the reviewer for this insightful comment.

At microwave frequencies, AlSi10Mg shows a very low metallic loss that can be ignored, just like other conventional metals, e.g., Au, Ag, and Cu. To confirm this, we designed a structure supporting surface waves. We fabricated two samples made of AlSi10Mg (see Fig. R1a) and copper (see Fig. R1b, copper is usually regarded approximately as a perfect electric conductor (PEC) at low frequencies), respectively. Using microwave measurement, we obtained the field distributions and the corresponding band structures of the surface waves (see Fig. R1c-e). We also compare our experimental results with simulated ones; in simulations, we set the metals as PEC. One can see that the measured results of the two metals are almost identical, and agree well with the simulated counterparts. All these results demonstrate that at microwave frequencies, AlSi10Mg has a negligible metallic loss, similar to the copper, and can be regarded approximately as PEC.

Fig. R1 a,b, Fabricated samples made of AlSi10Mg and Cu, respectively. The red star represents the source location. **c,d**, Measured dispersions of AlSi10Mg sample and Cu sample, respectively. The green dots display the simulated dispersion of a PEC structure, and the blue line shows the light line. **e**, Simulated and measured field distributions at different frequencies. The left, middle, and right columns represent the results of the structures made of PEC, AlSi10Mg, and Cu, respectively

Accordingly, we have included the above discussion as a new section in Supplementary Information Note 10 entitled “Microwave properties of AlSi10Mg”.

We have also added a discussion on line 268, page 11 of the main text, which reads as

“Notably, the AlSi10Mg at microwave frequencies has a negligible metallic loss, which can be regarded approximately as PEC. (see Supplementary Information Note 10)”.

2nd Reviewer -- Comment 2:

For COMSOL simulation, has the size of the system been varied to confirm convergence?

Response from Authors:

Yes.

For the surface dispersion, we have changed the supercell size from 4 to 15 unit cells, and the simulated results are displayed in Fig. R2a-d. One can see that the surface dispersion is convergent when the size reaches 8 unit cells.

Fig. R2 a-d, The simulated surface dispersion when the supercell size changes from 4 to 15 unit cells.

For the hinge dispersion, we change the supercell size from 3×3 to 11×11 unit cells, and the simulated results are displayed in Fig. R3a-d. One can see that the hinge dispersion is convergent when the size reaches 8×8 unit cells.

Fig. R3 a-d, The simulated hinge dispersion when the supercell size changes from 3×3 to 11×11 unit cells.

Following the reviewer’s comment, we have included the above discussion as a new section in Supplementary Information Note 11 entitled “Convergence of simulation results”.

We have also added a discussion on line 264, page 11 of the main text, which reads as

“In simulation, the size of the system has been varied to confirm convergence (see Supplementary Information Note 11).”

2nd Reviewer -- Comment 3:

A few minor issues. Headings can be helpful in the main article to help readers follow the logic. It is also helpful to special the sections when referring to the supplementary information. Also, there are a few typos including integratable (integrable) and tirival (trivial).

Response from Authors:

We thank the reviewer for the constructive comments.

Following the reviewer's comment, we have added the headings in the manuscript, as below

1. Paragraphs 1 to 3: Introduction
2. Paragraphs 4 to 7: Design of the 3D real higher-order Weyl PhC
3. Paragraphs 8 to 10: Experimental demonstration of the real higher-order Weyl points
4. Paragraphs 11: Discussion

We have indicated the sections when referring to the supplementary information in the manuscript.

The details are displayed as follows:

1. In line 89, page 3, we have added "see Supplementary Information Note 3".
2. In line 123, page 4, we have added "see Supplementary Information Note 8"
3. In line 144, page 4, we have added "see Supplementary Information Note 8"
4. In line 149, page 5, we have added "see Supplementary Information Note 9"
5. In line 156, page 5, we have added "see Supplementary Information Note 3"
6. In line 171, page 6, we have added "see Supplementary Information Note 10"
7. In line 221, page 9, we have added "see Supplementary Information Note 1"

We have also corrected all typos in the manuscript, the details are displayed as follows:

1. In line 95, page 3 and line 249, page 10, the typo "integratable" has been changed to "integrable".
2. In line 145, page 4, the typo "nontrival" has been changed to "nontrivial".

REVIEWERS' COMMENTS

Reviewer #1 (Remarks to the Author):

I think the authors have answered my questions satisfactorily. I recommend the publication of this work.

Reviewer #2 (Remarks to the Author):

The authors have satisfied my prior concerns with either clarification, or additional plots, and I am satisfied with the revised manuscript.

Response Letter to Reviewers

GENERAL COMMENTS FROM 1st REVIEWER:

I think the authors have answered my questions satisfactorily. I recommend the publication of this work.

Response from Authors:

We thank the reviewer for the recommendation for the publication of this work.

GENERAL COMMENTS FROM 2nd REVIEWER:

The authors have satisfied my prior concerns with either clarification, or additional plots, and I am satisfied with the revised manuscript.

Response from Authors:

We thank the reviewer for the recommendation for the publication of this work.